# Understanding *WT1* Alterations and Expression Profiles in Hematological Malignancies

**DOI:** 10.3390/cancers15133491

**Published:** 2023-07-04

**Authors:** Naghmeh Niktoreh, Lisa Weber, Christiane Walter, Mahshad Karimifard, Lina Marie Hoffmeister, Hannah Breiter, Aniththa Thivakaran, Maren Soldierer, Hans Günther Drexler, Heiner Schaal, Stephanie Sendker, Dirk Reinhardt, Markus Schneider, Helmut Hanenberg

**Affiliations:** 1Department of Pediatrics III, University Children’s Hospital Essen, University of Duisburg-Essen, 45147 Essen, Germanydirk.reinhardt@uk-essen.de (D.R.);; 2Faculty of Life Sciences, Technical University of Braunschweig, 38106 Braunschweig, Germany; 3Institute of Virology, University Hospital Düsseldorf, Medical Faculty, Heinrich Heine University, 40225 Düsseldorf, Germany; schaal@uni-duesseldorf.de; 4Department of Otorhinolaryngology, Head & Neck Surgery, University Hospital Düsseldorf, Heinrich Heine University, 40225 Düsseldorf, Germany

**Keywords:** *WT1*, *Wilms tumor one* gene, acute myeloid leukemia, acute lymphoblastic leukemia, nonsense-mediated mRNA decay

## Abstract

**Simple Summary:**

The transcription factor Wilms tumor one (WT1) is highly expressed in malignant cells of patients with hematological malignancies, with a considerable percentage of patients harboring deactivating mutations in this gene. To date, the extent of WT1’s contribution to leukemogenesis is still unclear and this, at least in part, results from the complexity of WT1 regulation and is further complicated by the lack of an appropriate model system to study *WT1* isoform formation under settings closer to reality. In the present study, we describe a set of leukemia and lymphoma cell lines, which are carefully characterized for WT1’s unique properties and thoroughly screened for sequence alterations in *WT1* and other genes important for leukemogenesis. We studied the formation of endogenous truncated WT1 and developed a model system for the study of WT1 major isoforms, thus significantly contributing to the field of WT1 research.

**Abstract:**

*WT1* is a true chameleon, both acting as an oncogene and tumor suppressor. As its exact role in leukemogenesis is still ambiguous, research with model systems representing natural conditions surrounding the genetic alterations in WT1 is necessary. In a cohort of 59 leukemia/lymphoma cell lines, we showed aberrant expression for *WT1* mRNA, which does not always translate into protein levels. We also analyzed the expression pattern of the four major WT1 protein isoforms in the cell lines and primary AML blasts with/without *WT1* mutations and demonstrated that the presence of mutations does not influence these patterns. By introduction of key intronic and exonic sequences of *WT1* into a lentiviral expression vector, we developed a unique tool that can stably overexpress the four *WT1* isoforms at their naturally occurring tissue-dependent ratio. To develop better cellular model systems for WT1, we sequenced large parts of its gene locus and also other important myeloid risk factor genes and revealed previously unknown alterations. Functionally, inhibition of the nonsense-mediated mRNA decay machinery revealed that under natural conditions, the mutated WT1 alleles go through a robust degradation. These results offer new insights and model systems regarding the characteristics of WT1 in leukemia and lymphoma.

## 1. Introduction

The *Wilms tumor one* (*WT1*) gene encodes a transcription factor deeply involved in the regulation of a set of genes accountable for the development of healthy hematopoiesis and the normal development of the genitourinary tract [1]. The role of *WT1* in the evolution of these tissues has proven to be crucial, since *Wt1*^−/−^ animals do not survive the embryonic stage, and deletion of *Wt1* in an inducible model of young adult mice led to fatal consequences such as severe glomerulosclerosis and impaired hematopoiesis with a diminished erythropoiesis [2,3]. Mechanistically, the WT1 protein exerts its regulatory effects via an amino-terminal transactivation domain encoded by exons 1 to 4 of the gene and a carboxyl terminal DNA binding domain consisted of four zinc fingers (ZFs) translated from exons 6 to 10 [4]. Alternative mRNA splicing adds, in addition to an alternative CTG start codon and post-translational modifications, another layer of complexity for understanding the role of WT1 in normal and malignant cells. The alternative splicing leads to the generation of several WT1 isoforms, with the four major ones resulting from two main alternative splicing events [5]. The first splicing event leads to inclusion or exclusion of 17 amino acids between the transactivation and DNA binding domains via inclusion or exclusion of exon 5, and the second event to inclusion or exclusion of three amino acids, lysin, threonine and serin (KTS), by two alternative splice donor sites at the end of exon 9. These isoforms, which hold unique properties and in part stimulate exclusive target genes, are expressed at constant but tissue-dependent ratios [6,7,8,9].

Although *WT1* was first identified via its genetic mutations in Wilms tumor (WT), a childhood malignancy of the kidneys, aberrant expression patterns of this gene have been observed in many types of solid cancers and even in a subset of WTs [10]. An aberrant WT1 expression was also observed in leukemic blasts of patients with acute lymphoblastic leukemia (ALL) of B- or T-lineages, acute myeloid leukemia (AML), chronic myeloid leukemia (CML) in blast crisis phase and myelodysplastic syndrome (MDS) [11]. Many efforts have been undertaken to define the exact role of WT1 in leukemogenesis; however, their collective results still remain elusive. Evidence suggests that the expression of WT1 is necessary for proliferation of leukemic blasts, but its overexpression in hematopoietic progenitor cells did not provide any proliferation advantage for these cells nor did it promote a malignant transformation [11,12,13]. In a transgenic mice model, co-expression of WT1 and the AML-ETO fusion protein led to a rapid onset of murine leukemia; however, neither WT1 nor AML-ETO alone sufficed here [14]. These results were contradicted in the subsequent work, where addition of WT1 was not necessary for leukemic transformation of mice already expressing two other oncogenic transgene combinations [15]. In addition, studies analyzing the expression pattern of WT1 isoforms have claimed that dysregulation in the isoform ratios contributes to leukemogenesis and even treatment outcomes in children and adults with AML [16,17,18]. However, research so far has been limited to overexpressing one or two *WT1* isoforms in separate samples and, to our knowledge, no model system mimicking *WT1*’s naturally occurring isoform ratios has been reported to date.

Somatic mutations in *WT1* are detected in considerable number of patients with acute leukemia and often co-occur with other genetic alterations, for instance internal tandem duplications of the *fms related receptor tyrosine kinase 3* (*FLT3-ITD*) gene or mutations in *CCAAT/enhancer-binding protein alpha* (*CEBPA*) in adult AML [19,20]. We and others have detected *WT1* mutations in 8–14% of children and adolescences with AML and observed a co-occurrence with *FLT3-ITD* mutations and *nucleoporin 98* fusion transcripts with *nuclear receptor binding SET domain protein 1* (*NUP98::NSD1*) [21,22]. The majority of the leukemia-associated *WT1* mutations are heterozygous events in exons 7 and 9, introducing small deletions, insertions or point mutations, which usually lead to nonsense-mediated mRNA decay or, if any protein is made, a premature stop with elimination of all or the last two ZF domains [11]. Importantly, forced expression of these altered gene products in hematopoietic CD34^+^ progenitor cells pointed to a possible role in leukemogenesis via a pro-proliferation effect [23].

Therefore, the two contradictory roles of *WT1* in leukemia, i.e., being an oncogene when aberrantly expressed and a tumor suppressor gene when mutated, has made this gene a subject for numerous research projects. However, the complexity of its function, its multiple isoforms and lack of appropriate model systems taking all features of this genes into consideration have left substantial gaps in our knowledge. In the present study, we sought to solve some of these issues by offering a directory of lymphoma and leukemia cell lines, well characterized for expression of *WT1* mRNA and protein and extensively screened for *WT1* alterations across its entire coding and splicing regions. We also developed a novel model system, which expresses the four major WT1 protein isoforms at their naturally occurring tissue-dependent ratios and therefore is well suited for studying the effects of *WT1* genetic alterations on the isoform formation. Together with the information provided in this manuscript, this novel tool will support future research closer to the reality of *WT1* in leukemia.

## 2. Materials and Methods

### 2.1. Cell Culture and Preparation of Genomic Material

A total of 31 cell lines listed in Appendix A were originally obtained from the German collection of microorganisms and cell culture (DSMZ, Braunschweig, Germany). These cells were cultivated using standard cell culture conditions as described on the DSMZ website. For cultivation of the KASUMI-6 cell line, 10 ng/mL granulocyte-macrophage colony-stimulating factor (GM-CSF) was added to the culture medium. To perform the experiments within the scope of this manuscript, an additional 28 cell line pellets were kindly provided by the DSMZ. Genomic DNA was extracted using QIAamp^®^ DNA Blood Mini Kit (QIAGEN, Hilden, Germany) according to manufacturer’s instructions. Total messenger RNA (mRNA) was extracted using QIAamp^®^ RNA Blood Mini Kit (QIAGEN, Hilden, Germany) according to manufacturer’s instructions. The purity and concentration of the extracted DNA and RNA were determined with a NanoDrop-2000c (ThermoFisher Scientific Inc., Waltham, MA, USA). Prior to the experiments, all cell lines were authenticated via short tandem repeat (STR) DNA genotyping [24].

### 2.2. Patient Samples

Primary AML blasts were obtained from the material left over after performing standard diagnostic procedures of the AML-BFM reference laboratory as previously described [25,26]. All material were collected at time of first diagnosis of AML. From patients originally recruited in AML-BFM 04 trial (ClinicalTrials.gov Identifier: NCT00111345) or the AML-BFM 2012 registry and trial (EudraCT number: 2013-000018-39), 24 with *WT1* mutations were randomly selected and further characterized using the methods described below. The clinical trials were approved by ethics committees and institutional review boards of University Hospital Münster (for AML-BFM 04) and University Hospital Hannover (for AML-BFM 2012). Informed consent was provided by the patients or their legal guardians before enrollment of the patients.

### 2.3. Complementary DNA Synthesis and RT-PCR

Complementary DNA (cDNA) synthesis was carried out with 1 μg of total mRNA and a mixture of oligo (dT)18 and random hexamer primers provided by SuperScript^TM^ VILO cDNA Synthesis Kit following the manufacturer’s instructions (ThermoFisher Scientific Inc., Waltham, MA, USA). The expression of target genes was analyzed using the TaqMan^TM^ system and region-specific primers and probes for *WT1* and *ABL Proto-Oncogene 1* (*ABL1*) gene as described in Appendix A. The real-time PCR (RT-PCR) strategy included 50 amplification cycles, each with a duration of 15 s at 95 °C and one minute at 60 °C. The *WT1* cycle threshold (Ct) was normalized using the delta Ct method in relation to the expression of ABL1 (ΔCt) and HEK293 cells (ΔΔCt).

### 2.4. Intracellular Staining of WT1 Protein and Flow Cytometry

One day after splitting the cell cultures, 1 × 10^6^ cells were fixed with 4% paraformaldehyde at room temperature for 10 min and then permeabilized using 90% methanol for 30 min at −20 °C. After washing, staining of the cells with monoclonal antibodies against WT1 (ab89901, Abcam, Cambridge, UK) and its isotype control (ab172730, Abcam, Cambridge, UK) was performed for 30 min at room temperature. Secondary staining was done with a directly conjugated antibody (Goat anti-Rabbit, H + L, Alexa Flour 488, ThermoFisher Scientific Inc., MA, USA). As an additional control for successful permeabilization, cells were also stained for Ki67 (REA183-PE) and its isotype control (REA293-PE, both antibodies from Miltenyi Biotec, Bergisch Gladbach, Germany). The samples were measured on a MACSQuant^®^ Analyzer 10 (Miltenyi Biotec, Bergisch Gladbach, Germany) and the results analyzed using a FlowLogic^TM^ software, version 7.3 (Zürich, Switzerland).

### 2.5. Fragment Length Analysis

The cDNA was used as a template in a multiplex PCR amplification assay using fluorescent-labeled primers for *WT1* and *ABL1* as described elsewhere [18]. The PCR products were processed via capillary electrophoresis on a 3500 Genetic Analyzer (Applied Biosystem, Waltham, MA, USA) and the results were analyzed using GeneMapper^TM^ software, version 4.2 (Applied Biosystems, MA, USA).

### 2.6. Generating WT1 Expression Constructs

The cDNA coding for the longest *WT1* isoform, *WT1^+/+^,* was kindly provided by Dr. Brigitte Royer-Pokora (Department of Human Genetics, Heinrich-Heine-University, Düsseldorf, Germany). This isoform was cloned into a lentiviral expression vector downstream to a modified spleen focus-forming virus (SFFV) promoter and upstream to the *pac* gene encoding the puromycin N-acetyltransferase resistance gene separated by an internal ribosome entry site (IRES). Fragments encoding *WT1*’s major alternative splice sites were computationally designed via combining 150 base pairs (bps) of exon-intron boundaries from the 5′ and 3′ introns involved in the alternative splicing of *WT1* and virtually introduced into the *WT1*^+/+^ cDNA. To avoid generating novel splice donor and/or acceptor sites, a computational modeling of splicing outcomes was performed via the HEXplorer online tool [27,28] and the sites with the highest probability of introducing aberrant splicing were mutated. Final sequences were synthesized by GeneArt (ThermoFisher Scientific Inc., MA, USA) and cloned into the *WT1*^+/+^ construct, and the resulting vector from hereon is called *WT1*^universal^.

### 2.7. Production of Lentiviral Particles

The production of lentiviral particles in HEK293T cells was performed on 10 cm cell culture dishes using the vesicular stomatitis virus glycoprotein (VSV-G) pseudotype-system (6 μg per transfection) as previously described [29]. One day after transfection, the culture medium was exchanged with Iscove’s Modified Dulbecco’s Medium, substituted with 10% FBS, 1% P/S and 1% L-glutamine (Gibco, ThermoFisher Scientific Inc., MA, USA). After 24 h, viral particle-containing supernatants were harvested and filtered through a 0.45 μm filter. These supernatants were used fresh at a multiplicity of infection (MOI) ≤ 0.1 or stored at −80 °C until the time of use.

### 2.8. Next-Generation Sequencing

Per cell line, 50 ng DNA was used to create a library targeting 54 genes (listed in Appendix A) associated with myeloid malignancies using the TruSight^TM^ myeloid sequencing panel (Illumina, Inc., San Diego, CA, USA) as described previously [30,31]. Next-generation sequencing (NGS) was performed on a MiSeqDx™-System in the research mode with 2 × 150 bps paired end reads using the V2 reagents (Illumina, Inc., San Diego, CA, USA). Data analysis including base-calling and demultiplexing was performed on the MiSeqDX System and the MiSeq Reporter software, version 2.6 (Illumina, Inc., San Diego, CA, USA). The final variants were annotated, classified and filtered using Sophia DDM software, version 4-4.6-0 (Sophia Genetics, Rolle, Switzerland) and VariantStudio Software version 3.0 (Illumina, Inc., San Diego, CA, USA).

### 2.9. Sanger Sequencing

Depending on the GC content of different regions of the gene, 14 different PCRs were performed using primer pairs and strategies listed in Appendix A. The standard amplification protocol was carried out for 35 cycles, each containing 20 s of denaturation at 95 °C, followed by 20 s of annealing at 59° and 20 s of elongation at 72 °C. Touch-down amplification, however, was performed for 78 cycles, with a higher denaturation temperature of 98 °C followed by a gradual temperature reduction at the annealing phase from 80 °C to 56 °C in 20 s for 48 cycles and continuing for 20 s at 56 °C for the remaining 30 cycles. Similar to the standard protocol, the elongation was performed at 72 °C for all cycles. PCR products were subsequently purified by addition of 1 μL of exonuclease I and 2 μL of FastAP^TM^ thermosensitive alkaline phosphatase (both from ThermoFisher Scientific Inc., MA, USA) and the mixture was heated at 37 °C for 15 min and 85 °C for further 15 min in a thermocycler. The final products were sent to Microsynth Seqlab (Göttingen, Germany) for sanger sequencing and results were analyzed using the alignment tool of the SnapGene^®^ software, version 6.0.7 (Chicago, IL, USA).

## 3. Results

### 3.1. The Expression Profile of WT1 in Leukemia and Lymphoma Cells

In total, 34 hematologic cell lines were analyzed for expression of *WT1* mRNA using RT-PCR, 30 of which originate from malignant blasts of adult or pediatric patients with various FAB subtypes of AML or with CML in blast crisis, three cell lines were derived from blasts of adult or pediatric patients with ALL and one cell line, U-937, was generated from an adult patient with histiocytic lymphoma (also see Appendix A). The *WT1* mRNA expression levels of these cell lines were normalized to the one detected in HEK293 cells, a well-established human embryonic kidney cell line with endogenous expression of wild-type *WT1* alleles [32]. With a median of 6.8-fold (IQR: 9.7 fold) relative to HEK293 cells, most of the cell lines showed an increased expression of *WT1*, with the highest observed levels being 36-fold and 27-fold higher in FKH-1 (AML-M4) and NB-4 (AML-M3) cells, respectively (Figure 1A). Three cell lines showed expression levels lower than HEK293 cells (fold change to HEK293 < 1): NOMO-1 (AML-M5) with 0.4-fold, U-937 with 0.0003-fold and AP-1060 (AML-M3) with 0.0002-fold expression. No significant difference was detected in expression levels of different FAB subtypes of the AML cell lines or in the cell line wild type for *WT1* compared to the ones harboring *WT1* mutations (Appendix A). *WT1* mRNA expression was higher in cell lines derived from adult and young adults, compared to the ones isolated from pediatric patients (*p*-value: 0.032, Appendix A).

Next, we analyzed the expression of WT1 protein via intracellular staining and flow cytometry, using *WT1*^+/+^ lentivirally transduced U-937 cells (U-937^WT1+/+^) as positive control for the staining procedure (Figure 1B). In total, 30 cell lines were analyzed, with 23 being isolated from patients with AML or CML in blast crisis, four from patients with lymphoma and three from ALL patients (Appendix A). Matched RT-PCR data were available for 18 of these cell lines (Appendix A and Figure 1). WT1 protein expression was detectable in most of the analyzed cell lines, with NB-4 showing the highest expression levels for endogenous WT1 (MFI WT1/isotype: 6) and U-937 being the cell line with the lowest (almost no) expression of WT1 (MFI WT1/isotype: 0.2, Figure 1C). Except for four cell lines, namely BV-173 (CML), K-562 (CML), M-07e (AML-M7) and NALM-1 (CML), the WT1 protein expression levels were comparable to the mRNA levels (Appendix A).

### 3.2. WT1 Isoform Expression Pattern in Leukemic Cell Lines and Primary AML Blasts

To define isoform expression profile of *WT1* across different cell lines, we amplified the *WT1* cDNA with a primer pair binding to the fourth and the tenth exons of the gene, which are located upstream and downstream of the major alternative splice sites [18]. This resulted in generation of PCR products containing fragments with different lengths, each representing a distinct *WT1* mRNA isoform. The smallest fragment (533 bps) represented the smallest *WT1* isoform lacking the entire exon 5 as well as KTS at the end of exon 9 (*WT1*^−/−^). The second smallest isoform (*WT1*^−/+^) was detected at 542 bps and the two larger isoforms, *WT1*^+/−^ and *WT1^+/+^,* had sizes of 584 and 593 bps, respectively. Using this method, 27 cell lines were analyzed, 23 of which were derived from patients with AML or CML, three from patients with ALL and one from a patient with histiocytic lymphoma (Appendix A). As delineated in Figure 2A, the *WT1* isoforms were expressed at similar ratios across different cell lines with the longest isoform, *WT1^+/+^,* being the dominant transcript. This cohort contained four cell lines with an altered *WT1* allele, all harboring a heterozygous mutation (CTS, KASUMI-6, U-937 and MOLM-1, also see Appendix A). CTS, KASUMI-6 and MOLM-1 expressed *WT1* isoforms at ratios comparable to *WT1* wild-type cell lines, and in U-937, in line with the mRNA expression results (Figure 1A)*, WT1* isoform expression was not detectable. To investigate the effect of *WT1* sequence alterations on the isoform expression profile in a bigger sample size, 24 primary AML blasts from children and adolescents at initial disease manifestation with at least one *WT1* mutation were analyzed (Table 1). This selected patient cohort had a median age of 12 years (range: 1–17 years); the blasts predominantly belonged to the AML subtypes M1/M2 and most commonly carried the often co-occurring mutation *FLT3*-ITD (Appendix A). Therefore, the patient cohort analyzed here was similar to our recently published cohort of 353 patients, 48 of which harbored *WT1* mutations [21]. Although one third of the leukemic blasts of the patients had compound heterozygous *WT1* mutations (Table 1), the mRNA isoform expression profile in the mutated AML blasts still resembled the one detected in the *WT1* wild-type leukemic cell lines and also in the 16 primary AML blasts with monoallelic WT1 mutations (Figure 2). Therefore, we could not detect any significant differences in the relative mRNA expression of *WT1* isoforms between leukemic cells expressing one or two mutated *WT1* alleles compared to the cells expressing the *WT1* wild type.

To develop a model system, enabling replicating the tissue-dependent endogenous isoform expression profile of this gene shown in Figure 2, we designed a lentiviral expression construct (*WT1^universal^*), where we inserted the genomic 5′ and 3′ sequences of introns 4, 5 and 9 into the *WT1* cDNA, assuming that they contain additional intronic splicing regulatory elements for the recognition of the alternative splice sites, thus potentially enabling the major alternative isoforms to be generated. We then, at low MOIs (≤0.1), transduced U-937 cells, one of the cell lines with almost zero expression of endogenous *WT1*, with this construct and also with a construct that just expressed the open reading frame for *WT1*^+/+^ as positive control (Figure 3A). Subsequently, the total mRNA extracted from puromycin selected U-937 cells was subjected to fragment length analysis. As delineated in Figure 3B,C, the *WT1^universal^* construct induced the expression of all four *WT1* isoforms in ratios resembling the ones expressed in other leukemic cell lines and primary AML blasts.

### 3.3. Screening for WT1 Mutations

To gain a comprehensive overview regarding the status of genetic alterations in our cell lines, we performed a NGS analysis using the TruSight^TM^ myeloid sequencing panel (Illumina, Inc., San Diego, CA, USA). This panel analyses mutational hot spots in 54 genes essential for leukemogenesis in AML (Appendix A) [31]. From 34 analyzed cell lines, 32 were originated from patients with AML or CML in blast crisis and two were previously isolated from patients with lymphoma (Appendix A). In total, 104 variants were discovered and validated via the pipeline described in Appendix A, taking factors such as read depth and variant frequency into consideration [30]. From 96 validated variants, *TP53* was the most frequently altered gene in our data set with 16 alterations in 15 cell lines. This was followed by *ASXL1* (11 in 10), *NRAS* (8 in 8), *FLT3* (7 in 7), *TET2* (7 in 5), *KRAS* and *EZH2* (both 5 in 5) and finally, *RUNX1* and *WT1* both with 4 alterations in 4 cell lines (Figure 4, Appendix A). From the other 45 genes analyzed, 19 were found to be altered in either one or two cell lines (Figure 4) and no genomic sequence alterations were detected in the remaining 26 genes (Appendix A). To confirm variants status (previously reported vs. newly described) and consequence (benign vs. pathogenic or unknown), different databases including Cancer Cell Line Encyclopedia (CCLE, Broad Institute, Cambridge MA, USA), dbSNP and ClinVar (NCBI—National Institutes of Health (NIH, Bethesda MD, USA)), COSMIC (Wellcome Sanger Institute, Hinxton, UK), Cellosaurus (Swiss Institute of Bioinformatics, Lausanne, Switzerland) and LOVD (Leiden Open Variation Database, Leiden University Medical Center, Leiden, Netherlands) were used to search for information describing the variants detected in this data set. As delineated in Appendix A and Appendix A, 17 out of 96 variants were not previously reported. Among these variants, 9 detected in *TET2*, *BCOR*, *EZH2*, *STAG2* and *RUNX1* had a high probably of pathogenicity, while 8 variants detected in *TET2, EZH2, CEBPa*, *DNMT3A* and *ATRX* were considered as variants of uncertain significance (VUS).

Using NGS, we identified four *WT1* alterations in four leukemic cells lines, namely CTS, KASUMI-6, U-937 and MOLM-1, all previously reported in these cells (Appendix A). CTS and KASUMI-6 harbored an identical mutation, a deletion of two base pairs in exon 7, leading to a frameshift and introduction of a premature termination codon. In U-937 and MOLM-1 cells, single bp alterations were detected: a nonsense mutation (p.R369Ter) in exon 7 of U-937 and a missense mutation (p.R462Q) in exon 9 of MOLM-1. While the first three changes in *WT1* are known to be pathogenic, the missense alteration in MOLM-1 cells is considered to be a VUS with a high probability of pathogenicity (LOVD data entry: WT1_000144). Since the TruSight^TM^ myeloid panel only analyzes the mutational hotspots in exons 7 and 9 of *WT1*, we additionally performed a comprehensive Sanger sequencing for the complete coding sequence, 50 bps from the intronic splice donor and acceptor sequences, i.e., 50 bps from the 5′ and 3′ end of the respective intron, as well as the 5′- and 3′UTRs (including the upstream region of 5′UTR). In total, 20 cell lines were sequenced; NGS results were also available for 18 out of these 20 (Appendix A). As described in Table 2, in addition to confirming all four variants detected via NGS, Sanger sequencing revealed a missense alteration (p.R495Q) in exon 10 of ME-1 cells, a location which is not covered by TruSight^TM^ myeloid panel. This alteration has been previously reported in this cell line (CCLE data entry: ACH-000439); however, its pathogenicity remains unknown. Moreover, other benign or likely benign variants were detected in exons and intronic areas and several VUS were discovered in the intronic part of the splice sites or in the 5′- and 3′UTRs (Table 2).

### 3.4. Expression of WT1 Mutated Alleles

To determine the functional consequence of mutations in *WT1*, we inhibited the nonsense-mediated mRNA decay machinery in the three cell lines harboring heterozygous nonsense mutations leading to a premature stop (CTS, KASUMI-6, U-937). As delineated in Figure 5A, treatment of these cell lines with cycloheximide (CHX), an inhibitor of translation elongation in eukaryotes [33], for six hours increased the *WT1* mRNA expression in a dose dependent manner in CTS and KASUMI-6 cells. The fragment length analysis revealed that the inhibition of nonsense-mediated mRNA decay is the main cause for the increased expression of the mutated *WT1* alleles (Figure 5B,C). Finally, although we only identified one nonsense mutation with premature termination codon via NGS as well as sanger sequencing in U-937 cells, expression levels of *WT1* still remained low/undetectable after treatment of this cell line with CHX.

## 4. Discussion

Defining the exact role of WT1 in leukemogenesis serves a dilemma, since its aberrant expression is detected in vast majority of leukemic blasts, even in those that harbor simultaneous mutations on the second allele [8]. Clinically, although a consensus regarding the prognosis of WT1 expression level in newly diagnosed acute leukemia is still lacking, the WT1 expression has been successfully used as a marker of minimal residual disease, reliably predicting relapse after (induction of) chemotherapy or after hematopoietic stem cell transplantation [34,35,36]. In addition, attempts to investigate the prognostic impact of *WT1* mutations in acute leukemia remain nonconcurrent: several studies recognized *WT1* mutations as independent predictors of poor treatment outcomes [19,20,37,38,39,40], while others described the effect of *WT1* mutations to be dependent on other co-alterations [20,21]. A better understanding of *WT1*’s role in leukemogenesis is needed to solve these discrepancies.

In this manuscript, we have generated a comprehensive directory of leukemia and lymphoma cell lines well defined for characteristics of *WT1* genomic alterations as well as mRNA and protein expression. We show that *WT1* is highly expressed in most of the analyzed cell lines; however, almost no *WT1* mRNA could be detected in AP-1060 and U-937 cells. In AP-1060, an acute promyelocytic leukemia cell line derived form an adult patient, this can be explained by the fact that these cells harbor a homozygous deletion in the 11p13 locus, leading to bi-allelic loss of *WT1* expression [41]. In the histiocytic lymphoma cell line U-937, we could only confirm the known nonsense mutation in exon 7 on one allele. Although other genetic/cytogenetic alterations surrounding *WT1* locus have neither been reported nor were found by us, we show here that the expression of *WT1* even after inhibition of nonsense-mediated mRNA decay remains undetectable in U-937 cells, pointing to epigenetic regulations that may have resulted in *WT1* silencing. In addition, we observed higher *WT1* expression in cell lines derived from adults compared to the ones derived from pediatric patients. However, since an association between *WT1* expression and age has not been reported in large patient cohorts [11], this finding might be a consequence of the small sample size in the current study.

Expression of WT1 protein is often measured via immunohistochemistry in solid tumors [42]. Evidence from a meta-analysis of 3620 patients suggested that while both, the WT1 mRNA as well as protein levels were independent predictors of worse treatment outcomes, the latter parameter had a stronger prognostic power [42]. Contrary to solid tumors, WT1 protein expression is not routinely analyzed in hematologic malignancies; thus, information regarding its prognostic value or its correlation with the *WT1* mRNA levels are not generally available. In the current study, we also measured the expression of WT1 protein using flow cytometry on permeabilized cells and showed that the expression of mRNA and protein were congruent in most of the cell lines. However, discrepancies between mRNA and protein expression in general are not uncommon observations [43,44] and can additionally be influenced by post-translational regulatory modifications involved in cell development and homeostasis [45,46]. A comprehensive comparison of proteomics and mRNA micro arrays in hematopoietic cell lines in steady and dynamic states showed that often a moderate correlation exists between mRNA and protein expression levels [47].

The equilibrium of *WT1* isoform ratio is a key factor of the functions of this gene in different tissues, as evident for example by Frasier syndrome where a bi-allelic point mutation in the splice donor of exon 9 results in disturbance of *WT1*^+/+^/*WT1*^+/−^ and *WT1*^−/+^/*WT1*^−/−^ ratios, leading to major complications such as male pseudo-hermaphroditism and progressive glomerulopathy [48,49]. According to our results presented here and data published previously by others [16,17,18], *WT1* isoforms are expressed in malignant myeloid blasts derived from patients with AML and CML at constant ratios with a predominant expression observed for *WT1^+/+^.* We found the ratios for the four major *WT1* isoforms to be quite conserved across different cell lines and primary AML blasts with and without *WT1* mutations, as the splicing patterns are the result of the *WT1* gene structure and mutations in factors of the spliceosome are quite rare and only occur in a monoallelic constellation in the malignant cells analyzed. Therefore, the splicing patterns are likely a reflection of splicing regulatory elements at the normal exon-intron boundaries in *WT1*. Although other studies have reported similar expression pattern of *WT1* isoforms within leukemic blasts, the isoform expression ratios in AML blasts were shown to be disproportional to the ratios observed in healthy CD34^+^ hematopoietic progenitors isolated from mobilized clonogenic cells in peripheral blood [16,17] or umbilical cord blood cells [18]. In a cohort of 112 children and adults with myeloid malignancies, Luna et al. detected lower *WT1*^+/+^ expression levels in AML blasts compared to CD34^+^ cord blood cells [18]. The authors related the high *WT1* expression in patients to the cumulative expression level of the other three isoforms and also suggested a direct correlation between dismal outcomes and the *WT1*^+/−^ expression levels [18]. So far, this interesting concept is yet to be confirmed by others.

Replicating the WT1 splicing patterns under controlled experimental settings requires a model system that honors the endogenous regulatory mechanisms involved in the process of *WT1* isoform expression. With inclusion of *WT1*’s major alternative splice sites into a *WT1* expression vector, we have developed such a system in the presented work and show that this novel construct is capable of expressing all four *WT1* isoforms at the expected ratios and levels in cells. Our results confirmed that the small regions of introns, up- and downstream of the exon boundaries hold the genomic information sufficient to induce splicing and alternative splicing events [50,51]. Introducing this construct in normal and malignant cells of different origin will facilitate to better understand formation of the four isoforms per se and also allows to systematically introduce or delete factors in these cells that will influence the splicing pattern. Another approach to use this unique expression system would be to replace the strong SFFV promoter with weaker options, e.g., *WT1*’s endogenous or a regulatable promoter, and then to study the effect of WT1 overexpression on leukemogenesis, differentiation and chemotherapy resistance in normal CD34+ progenitor cells that additionally co-express other type I or type II mutations detected in AML blasts. The functionality of WT1 expression levels would be characterized in short-term assays in vitro or in vivo in immunodeficient mice. Importantly, the same approach can be used to introduce different *WT1* alterations, especially the variants of unknown significance, and investigate their cellular consequences. Functional mutation testing with our lentiviral overexpression system principally is also possible; however, the fact that NMD plays a major role in malignant blasts needs to be taken into account when investigating nonsense mutations (premature stops, indels) with potentially dominant negative effects.

Consistent to previous findings, even in an extended sequencing panel, we found almost all pathologic mutations of *WT1* to be located on exons 7 and 9 of the gene. The consequence of R495Q in ME-1 (exon 10) and intronic variants close to the splice sites discovered in the current study, need to be defined in future functional studies. Interestingly, the detected R368Ter in U-937 (exon 9) has been previously reported in patients with Denys–Drash syndrome, a rare condition with gonadal dysgenesis, nephropathy, and WT [52,53]. The mutations in *WT1* exon 7 in CTS and KASUMI-6 resembled common mutations in acute leukemia which lead to two presumed scenarios: either creation of truncated proteins with diminished or lost DNA binding capacity but intact 5′ transactivation, self-association and RNA-recognition domains or degradation by nonsense-mediated mRNA decay machinery and a possible subsequent haploinsufficiency. Many studies have investigated the pathogenesis of the first scenario and uncovered possible key aspects [23,54,55,56,57]. For example, when artificially overexpressed in cells with the wild-type WT1 isoforms, truncated WT1 lacking all ZFs (WT1-delZF) caused a significant reduction in transcriptional activation of wild-type WT1 and dramatic changes in its subnuclear localization, due to dimerization of truncated and wild-type WT1 proteins [54,55]. In more recent experiments, forced expression of WT1-delZF, for example, led to significant phenotypic changes in hematopoietic tissues [23,56,57]. In a study by Vidovic et al., overexpression of WT1-delZF in CD34^+^ hematopoietic cells caused a shift to a proliferative phenotype with an accentuated erythropoiesis in colony formation assays [23]. Gene expression analysis also showed a unique set of up- and down-regulated target genes in cells expressing WT1-delZF, which promoted erythroid maturation even in the absence of erythropoietin [23]. Although the exact mechanisms behind the function of WT1-delZF are still not completely clear, involvement in post-transcriptional mechanisms seem plausible since these truncated proteins cannot bind to DNA but still have an intact RNA-binding domain. Epigenetic dysregulation as a distinct DNA-independent mechanism could also be caused by the compromised interaction of WT1-delZF and ten-eleven translocation-2 (TET2), a catalyzer of oxidation with essential role in myelopoiesis [58,59,60,61,62]. As described by Rampal et al. and Wang et al., WT1-delZF cannot recruit TET2 to its target genes, thus blocks the IDH1/2-TET2-WT1 axis and subsequently leads to DNA hypermethylation [58,59,60].

The truncating mutations in *WT1* in the AML blasts could also result in expression of shortened WT1 proteins. However, to our knowledge, these mutated proteins have not been detected so far and the evidence stemming from the above-mentioned functional studies is an aftermath of transient transfection or permanent transduction experiments using strong viral promoters for the expression of cDNAs. As we showed here, mutated transcripts are detectable at higher levels only when inhibition of the nonsense-mediated mRNA decay machinery (e.g., via cycloheximide treatment) is performed. The significant increase in the mRNA expression of the mutated alleles strongly argues against the existence of functionally relevant levels of these mutant proteins. Although these are limited to the analyzed cell lines, similar results were also achieved in another study using emetine as an alternative inhibitor of eukaryote protein synthesis in primary AML blasts [63]. By inhibiting the nonsense-mediated mRNA decay machinery in primary AML blasts and the CTS cell line, Abbas et al. detected a “re-appearance” of *WT1* mutations in mRNA sequencing. Interestingly, while the wild-type WT1 from the unaffected alleles was abundant in all samples, the authors still could not detect the truncated WT1 proteins via Western blot [63].

As evident by the results of the current study and also many others, it is well established that the wild-type WT1 protein is highly expressed in acute leukemia, even in patients with *WT1* mutations [63,64]. Thus, it is tempting to speculate that only the expression of the wild-type allele contributes to leukemogenesis. Although the extent of this contribution still needs to be clarified, high expression of WT1 offers a great chance for developing targeted therapies, including immunological approaches. Several studies have used the wild-type or modified major histocompatibility complex (MHC) class I restricted WT1 peptides for generation of vaccination against WT1 overexpressing tumor cells and leukemic blasts [65]. Although phase I trials showed a tolerable and safe profile for these vaccinations, consequent phase II studies so far failed to prove their superiority to standard-of-care treatments [66,67,68,69,70,71]. Recently, with generation of several monoclonal antibodies against the WT1/MHC-I peptide complex expressed on the surface of the leukemic blasts [72,73,74], these antibodies mimicking the binding of a T-cell receptor (TCR) were further developed as bispecific T-cell engager antibodies (BITEs) or changed into a chimeric antigen receptor (CAR) format [75,76,77]. When used against acute leukemic blasts, both modalities showed a potent antileukemic effect in in vitro experiments and in patient-derived acute leukemia xenograft models [75,76,77]. Future research determining their *on-target/off-tumor* toxicities and enhancing their function on autologous or allogeneic immune effector cells are crucial next steps to be taken before introduction of these novel treatment modalities to the patients. However, due to the pronounced overexpression of WT1 in multiple tumors, developing *off-the-shelf* cellular therapies against WT1 are highly promising.

## 5. Conclusions

*WT1* is frequently mutated in leukemia and lymphoma blasts. Although we have shown here that the aftermath of some of these alterations still needs to be clarified, the majority of *WT1* mutations introduce premature termination codons. Consequently, these altered sequences are prominently subjected to nonsense-mediated mRNA decay and will be destroyed almost completely before translation into truncated proteins. However, regardless of the presence or absence of these mutations, our results indicate that the expression of *WT1* from the nonmutated second allele is high in the majority of hematological malignancies, showing a relatively consistent isoform pattern. The degree of the contribution of total *WT1* expression levels or the interrupted ratio of its isoforms to the survival outcomes in patients remains controversial across different studies. Nevertheless, the universal high expression of *WT1* alongside the novel achievements in generation of monoclonal antibodies against this protein expressed on the cell surface in the context of MHC class I molecules offer a unique chance to develop highly efficient, malignant cell-specific immunotherapies for pediatric and adult hematologic malignancies as well as for any other tumor entity with abundant *WT1* expression.

## Figures and Tables

**Figure 1 cancers-15-03491-f001:**
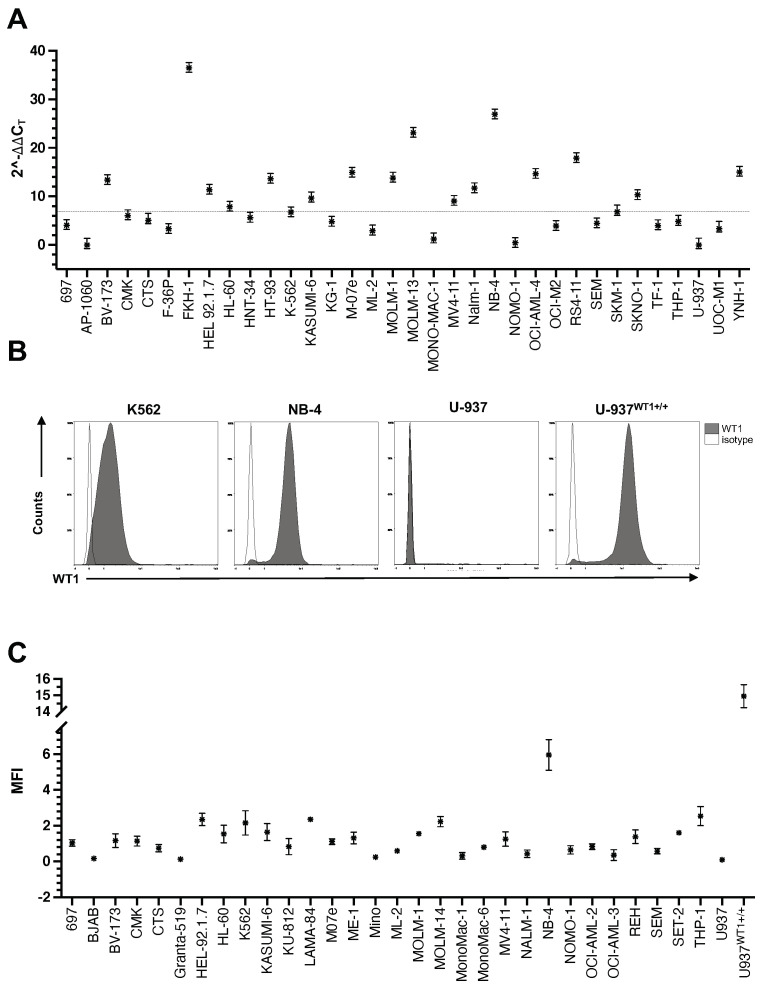
Expression of the *WT1* in leukemia. (**A**) Expression of *WT1* mRNA. Total mRNA was extracted from different cell lines and reverse-transcribed into cDNA. The cDNA was used as template for RT-PCR in combination with selective primer pairs for *WT1* exon 2 and *ABL1*. *WT1*’s cycle threshold was normalized to ABL1 (ΔCt) and HEK293 cells (ΔΔCt). The fold change expression normalized to that of detected in HEK293 cells was reported (2^−ΔΔCt^). The graph shows mean values ± SD of three biological replicates. Dashed line shows the median expression level in all cell lines (6.8-fold, IQR 9.7). (**B**,**C**) Expression of WT1 protein. One day after splitting cultures, 1 × 10^6^ cells per line were fixed and permeabilized using paraformaldehyde and methanol as described in the methods. The intracellular staining was performed for *WT1* antibody and its isotype control as well as Ki67 as a technical positive control (also see Appendix A). Panel (**B**) shows histograms of four example cell lines delineating the percentage of the WT1 positive cells. Refer to Figure 2B,C for detailed description regarding generation of U-937^WT1+/+^ cells. For panel (**C**), population’s MFI was normalized to that of expressed in isotype control via subtraction (ΔMFI = MFI_WT1_ − MFI_isotype_). The graph shows mean ± SE of two to three biological replicates, experiments from panel (**B**) repeated 2–3 times with similar results. Abbreviations: *WT1, Wilms tumor 1*; cDNA, complementary DNA; RT-PCR, real-time PCR; *ABL*, *Abelson tyrosine-protein kinase*; MFI, mean florescence intensity; *, represents 2^−ΔΔCt^ values in panel (**A**) and ΔMFI values in panel (**C**).

**Figure 2 cancers-15-03491-f002:**
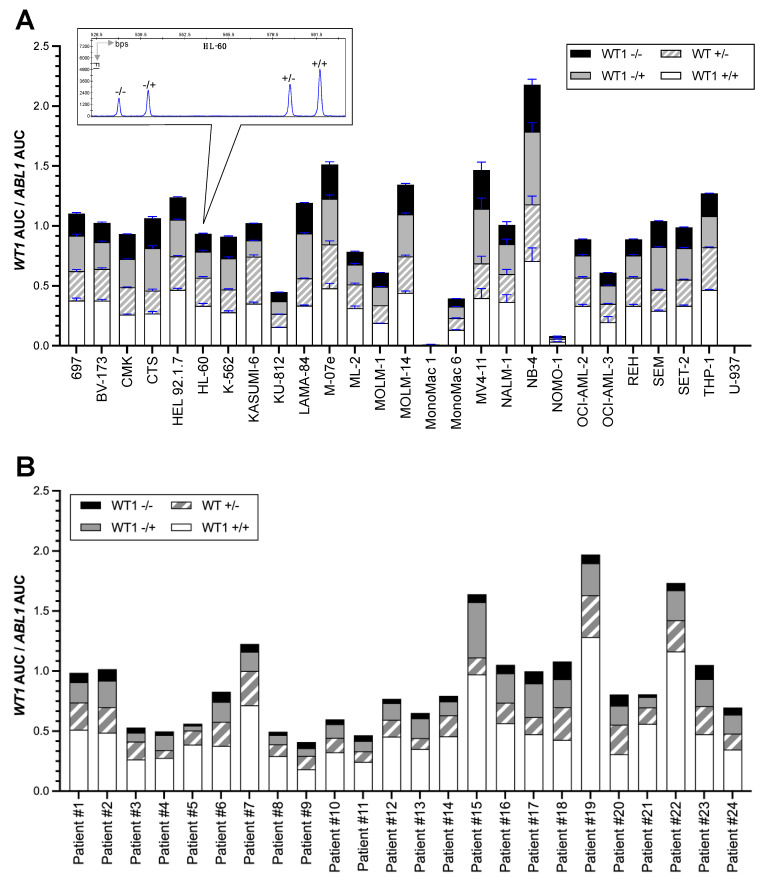
Profiling *WT1*’s isoform expression in acute leukemias. Total mRNA was extracted from the cell lines (Panel (**A**)) or primary AML blasts (Panel (**B**)) and reverse-transcribed into cDNA. Using the cDNA as templates, a multiplex PCR assay was performed for *WT1* and *ABL1* using fluorescent -labeled primers described in the methods. Fragments were separated based on their length by capillary electrophoresis. Areas under the curves of *WT1* peaks were normalized to *ABL1* by division AUC*_WT1_*/AUC*_ABL_*. Graphs in Panel (**A**), represent the mean ± SD from two biologic replicates. *WT1*^−/−^ represents *WT1* isoform lacking exon 5 as well as three amino acids, KTS at the end of exon 9. *WT1*^−/+^ represents the isoform lacking exon 5 but containing KTS. *WT1*^+/−^ contains exon 5 but lacks KTS. *WT1^+/+^,* represents the largest *WT1* isoform containing both exon 5 and KTS. Abbreviations: *WT1*, *Wilms tumor 1*; *ABL*, *Abelson tyrosine-protein kinase*; cDNA, complementary DNA; AUC, area under the curve.

**Figure 3 cancers-15-03491-f003:**
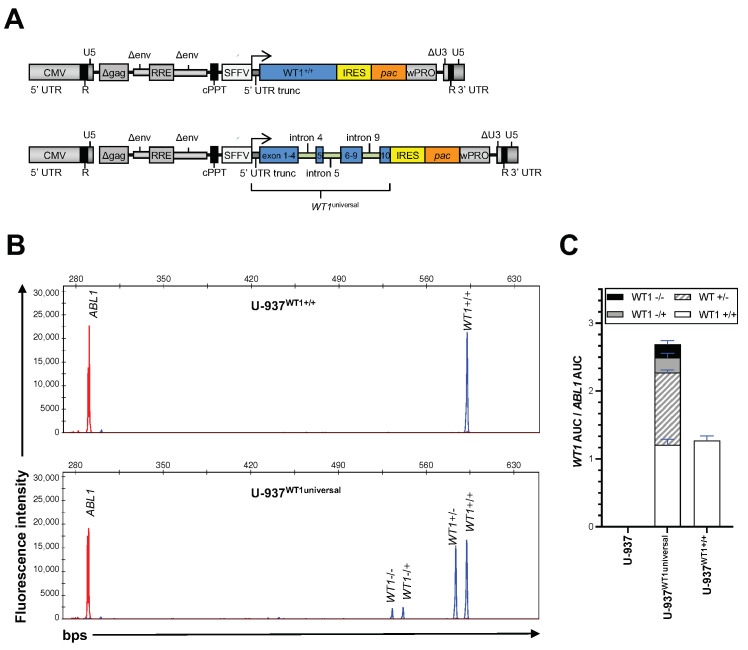
*WT1* isoform overexpression system. (**A**) Schematic presentation of lentiviral constructs expressing WT1 and *pac* gene encoding a puromycin N-acetyltransferase. (**B**) U-937 cells were transduced with WT1 expressing lentiviral particles pseudotyped with VSV-G. Five days after transduction, the cells were treated with 1 ug/mL Puromycin. Upon completion of selection of WT1 positive cells, total mRNA and cDNA was prepared from the cells and fragment-length analysis performed as described in panel (**A**). Different fragment curves shown represent different *WT1* isoform. (**C**) *WT1*’s AUC was normalized to that of expressed in *ABL1.* The graph shows mean ± SD of AUC*_WT1_*/AUC*_ABL_* from two biologic replicates. Experiments from panel (**B**) were repeated 2 times with similar results. *WT1*^−/−^ represents *WT1* isoform lacking exon 5 as well as three amino acids, KTS at the end of exon 9. *WT1*^−/+^ represents the isoform lacking exon 5 but containing KTS. *WT1*^+/−^ contains exon 5 but lacks KTS. *WT1^+/+^,* represents the largest *WT1* isoform containing both exon 5 and KTS. Abbreviations: *WT1, Wilms tumor 1*; cDNA, complementary DNA; *ABL*, *Abelson tyrosine-protein kinase*; AUC, area under the curve; VSV-G, vesicular stomatitis virus glycoprotein.

**Figure 4 cancers-15-03491-f004:**
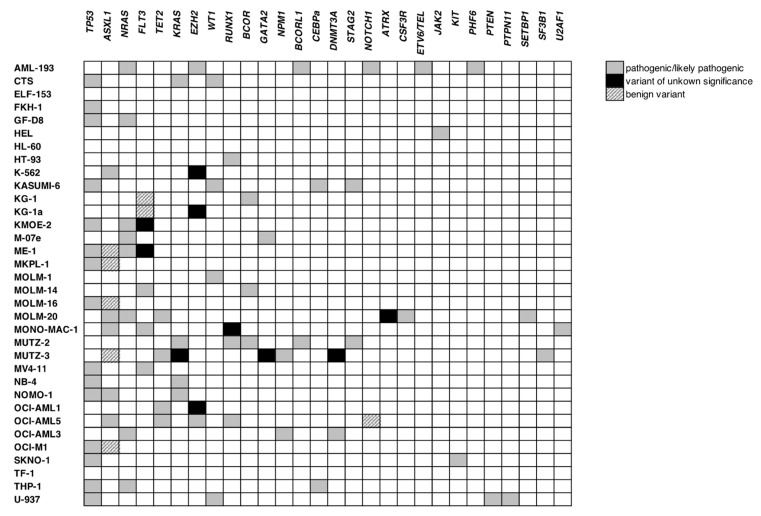
*WT1* alterations in different cell lines. An overview of NGS results. Variant validation and decision on their clinical relevance was performed as described in validation pipeline in Appendix A. Abbreviations: NGS, next-generation sequencing; VUS, variant of unknown significance.

**Figure 5 cancers-15-03491-f005:**
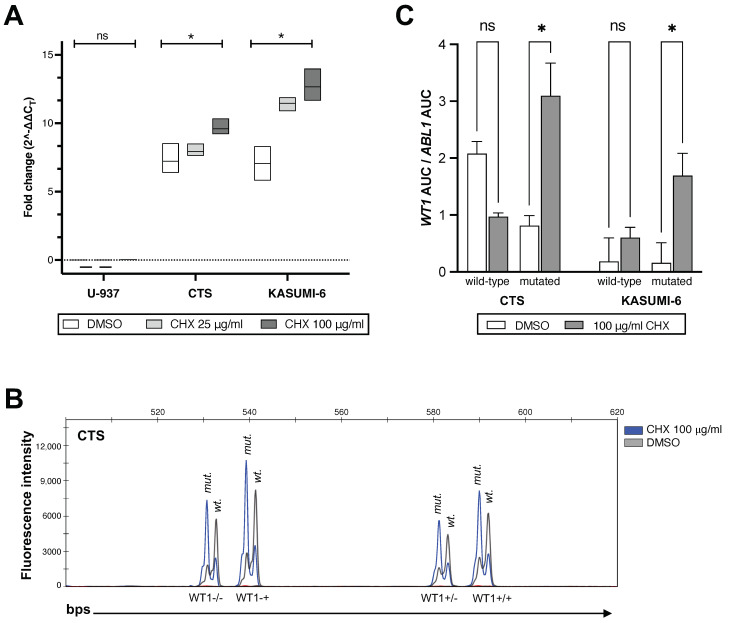
Inhibition of nonsense-mediated mRNA decay in *WT1* mutated cell lines. Cells were treated for six hours with either 25 or 100 μg/mL cycloheximide (CHX) or DMSO and then total mRNA was extracted. cDNA was used either for RT-PCR (Panel (**A**)) or fragment length analysis (Panels (**B**,**C**)). (**A**) The expression of *WT1* mRNA after CHX treatment was measured via RT-PCR as described in Figure 1 and the methods. *WT1*’s expression was normalized to *ABL1* (ΔCt) and HEK293 (ΔΔCt). The graph shows mean values ± SD of three biological replicates. (**B**) Overlay of the fragment length analysis showing the expression of *WT1*’s wild-type and mutated alleles on a capillary electrophoresis before and after treatment with CHX. Different *WT1* isoforms and alleles were separated based on their size. CTS cells harbor a deletion of two bps, thus each isoform shows two peaks with the mutated alleles being two bps shorter than the wild-type alleles. (**C**) Fragment length analysis was performed before and after CHX treatment and the curves representing *WT1*’s mutated and wild-type alleles were separated according to their size as described in panel (**B**). The AUC for each curve was normalized to *ABL1*’s AUC by division. The graph shows mean ± SD values from two biologic replicates. Abbreviations: CHX, Cycloheximide; *WT1, Wilms tumor 1*; wt., wild type; mut., mutated; DMSO, dimethyl sulfoxide; AUC, area under the curve; ns, not significant; *, stands for significant *p* values (<0.02) generated from two-way ANOVA tests.

**Table 1 cancers-15-03491-t001:** Characteristics of *WT1* mutations detected in primary blasts at initial diagnosis of AML.

P#	FAB	Variant 1				Variant 2				Loc.
		cDNA ^1^	AA ^2^	VF%	Exon	cDNA	AA	VF%	Exon	
1	M1	c.1057_1058insTA	p.Arg353LeufsTer6	45.8	7	c.1123dupA	p.Met375AsnfsTer9	44.5	7	biallelic
2	M4	c.1121delT	p.Phe374SerfsTer58	91.84	7					
3	M1	c.1133dupA	p.Tyr378fsTer1	46.56	7					
4	M4	c.1059dupT	p.Val354CysfsTer14	46.94	7	c.1082_1091dup TTGTACGGTC	p.Ala365CysfsTer6	40.65	7	biallelic
5	M1	c.1054_1055insT	p.Arg352LeufsTer16	67.2	7					
6	M0	c.1058_1059insGGTGCCGCTCG	p.Gly356LeufsTer6	48.49	7	c.1082_1091dup TTGTACGGTC	p.Ala365CysfsTer6	41.83	7	biallelic
7	M1	c.1078_1079insGCCGA	p.Thr360SerfsTer74	38.73	7	c.1084_1085insGC	p.Val362Glyfs* 71	52.92	7	biallelic
8	M1/M2	c.1087delCinsGGG	p.Arg363GlyfsTer70	52.11	7	c.1057delCinsGG	p.Arg353GlyfsTer15	42.78	7	biallelic
9	M1	c.1322_1332dupGAAAGTTCTCC	p.Arg445GlufsTer9	40.5	9	c.1323_1338dup AAAGTTCTCCCGGTCC	p.Asp447LysfsTer18	40.1	9	biallelic
10	M5	c.1068_1076del9ins GACGGTCGTTATTA	p.Val357ThrfsTer77	42.14	7					
11	M2/M4	c.1325dupA	p.Phe443ValfsTer17	13.87	9					
12	M4	c.1048-4_1056dup GCAGGATGTGCGA	p.Arg353AlafsTer19	30.25	7					
13	M2	c.1090_1091dupTC	p.Ala365ArgfsTer68	44.25	7					
14	M5	c.1079_1089dupCTCTTGTACGG	p.Ser364LeufsTer72	39.58	7					
15	M4	c.1080_1087dupTCTTGTAC	p.Arg363LeufsTer72	34.57	7					
16	M0	c.1077_1078ins TGTTTCTTCCGCCCAG	p.Thr360CysfsTer13	36.95	7					
17	M1	c.1086_1087insGAACTCTTGTA	p.Arg363GlufsTer73	30.15	7					
18	M4	c.1090_1091insAGGT	p.Ser364fsTer1	42.97	7					
19	M4eo	c.1056_1058delACGinsC	p.Arg353CysfsTer14	26.44	7	c.1089dupG	p.Ser364ValfsTer4	9.59	7	biallelic
20	M2	c.1079_1090del CTCTTGTACGGTinsTGGG	p.Thr360MetfsTer5	55.23	7					
21	M4	c.1074_1077dupCCCG	p.Thr360ProfsTer9	9.9	7					
22	M1/M2	c.1090_1093dupTCGG	p.Ala365ValfsTer4	6.94	7	c.1091dupC	p.Ala365GlyfsTer3	97.42	7	biallelic
23	M0	c.1087delCinsGG	p.Arg363GlyfsTer5	41.88	7					
24	M1/M2	c.1058delGinsCC	p.Arg353ProfsTer15	44.78	7					

Abbreviations. P#, patient number; FAB, French–American–British classification of AML; AA, amino acid sequence; VF, variant frequency; Loc., location; ins, insertion; del, deletion; dup, duplication; fs, frameshift; Ter, termination codon. ^1^ Nucleotide reference transcript ID: NM_024426.4. ^2^ Peptide reference sequence ID: NP_077744.3.

**Table 2 cancers-15-03491-t002:** Overview of *WT1* alterations detected via Sanger sequencing.

	Location	Variant (DNA) ^1^	Variant (AA) ^2^	Zygosity	Cell Line	Previous Reported Variant?		Variant Status in Analyzed Cell Lines
						Variant Ref.	Relevance	
**Exonic Variants**	exon1	c.330C > T	p.Pro110Pro	Heterozygous	CMK, FKH-1, HL-60, HAT-93, KASUMI-6, KG-1, ME-1, ML-2, MOLM-1, NOMO-1, SKNO-1, TF-1, THP-1, YNH-1	LOVD ^3^ (WT1_000137)	Benign	Current data set
		c.198G > T	p.Pro66Pro	Heterozygous	HL-60, HT-93, K-562, MOLM-1, M-07e, NB-4, SKNO-1, UOC-M1, U-937	dbSNP/ClinVar ^4^ (rs2234582)	Benign	Current data set
		c.594C > T	p.Asn98Asn	Heterozygous	HT-93, M-07e, NB-4, SKNO-1, UOC-M1	dbSNP/ClinVar (rs2234583)	Benign	Current data set
		c.294C > A	p.Gly98Gly	Heterozygous	KG-1	LOVD (WT1_000161)	Benign	Current data set
	exon 7	c.1202_1203delGA	p.Arg401fsTer3	Heterozygous	CTS	Not reported	Pathogenic	PMID 18591546
					KASUMI-6			CCLE ^5^ (ACH-000166)
		c.1107A > G	p.Arg369Arg	Heterozygous	CMK, HL-60, KG-1, ME-1, MOLM-1, NOMO-1, THP-1	dbSNP/ClinVar (rs16754)	Benign	Current data set
				Homozygous	HT-93, KASUMI-6, ML-2, SKNO-1, TF-1, YNH-1			
		c.1105C > T	p.Arg369Ter	Heterozygous	U-937	dbSNP/ClinVar (rs1423753702)	Pathogenic	Cellosaurus (CVCL_0007)
	exon 9	c.1385G > A	p.Arg462Gln	Heterozygous	MOLM-1	LOVD (WT1_000144)	Likely pathogenic	CCLE (ACH-001573)
	exon 10	c.1484G > A	p.Arg495Gln	Heterozygous	ME-1	Not reported	VUS	CCLE (ACH-000439)
**Intronic variants**	Upstream to 5′UTR	c.-883C > T	Intronic variant	Homozygous	All cell lines but CTS	Not reported	VUS	Current data set
	upstream to 5′UTR	c.-784T > C	Intronic variant	Homozygous	All cell lines but CTS	Not reported	VUS	Current data set
	upstream to 5′UTR	c.-654G > C	Intronic variant	Heterozygous	K-562	Not reported	VUS	Current data set
	5′UTR	c.-247T > C	Intronic variant	Homozygous	all cell lines but CTS	COSV60070502	VUS	Current data set
	intron 3	c.872 + 16G > A	Intronic variant	Heterozygous	HL-60, K-562, MOLM-1, M-07e, NB-4, UOC-M1	LOVD (WT1_000123) COSV60066980	Likely benign	Current data set
	intron 6	c.1099-9T > C	Intronic variant	Heterozygous	HL-60	LOVD (WT1_000117) COSV60072524	Likely benign	Current data set
	3′UTR	c.1554 + 88A > G	Intronic variant	Homozygous	UOC-M1	Not reported	VUS	Current data set

Abbreviations. AA, amino acid sequence; *WT1, Wilms tumor one*, LOVD, Leiden Open Variation Database; CCLE, cancer cell line encyclopedia; VUS, variant of unknown significance. ^1^ Nucleotide reference transcript ID: NM_024426.4; ^2^ Peptide reference sequence ID: NP_077744.3; ^3^ LOVD database accessed at https://databases.lovd.nl/shared/genes (accessed on 1 February 2023); ^4^ ClinVar database accessed at https://www.ncbi.nlm.nih.gov/clinvar (accessed on 1 February 2023); ^5^ CCLE database accessed at https://depmap.org (accessed on 1 February 2023).

## Data Availability

The cell line data presented in this study are available in main figures and tables as well as the Appendix A. Patient data presented in this study are available on request from the corresponding author and are not publicly available due to ethical considerations.

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
