# Peer review of "Understanding WT1 Alterations and Expression Profiles in Hematological Malignancies"

_cancers, 2023, doi:10.3390/cancers15133491_

Round 1

Reviewer 1 Report

nice work based on cell lines mainly

a new tool to study WT1 gene 

Overall we have a good presentation of a new tool not a new concept to study WT1 with a lot of cell lines. Cell lines data may be very different from primary tumors.

Author Response

We are thankful for the time that the reviewer has spent evaluating our manuscript and agree that the observations obtained from experiments using cell lines are not completely comparable with those performed with primary cells. However, considering the difficulties for genetic manipulation of primary leukemic blasts and the huge challenges for maintaining primary blasts in in vitro cultures, we believe that AML cell lines cannot be entirely eliminated from future experiments. Our manuscript offers a directory of well characterized cell lines that assists researchers to better select cell lines for their future experiments and thereby achieve optimal results. Nevertheless, we also included primary blasts in our analyses, therefore providing guidance where the similarities are obvious.

Reviewer 2 Report

Niktoreh et al. have addressed an issue of lake of comprehensive understanding of molecular heterogeneity of WT1 in leukemia and lymphoma. The lentiviral vector containing 4 different isoforms of WT1 is a valuable resource for studying the gene in hematological malignancies. The text is clearly written for the audience to follow the message.

There are several concerns as below for this manuscript. 

1. Abbreviations like PTC, NMD can be voided for easy reading

2. The term functional consequences gives misleading interpretation that the study demonstrates impact of WT1 on malignant transformation or suril or other cellular functions. The better term could be "Understanding mutation and gene expression interactions of  WT1 in adult? leukemia and lymphoma"

3. Give a brief description of the patients included in this study from trials AML-BFM 04 and AML-BFM 2012. Clarify why patient samples were obtained as part of this trials. Justify that there was no genetic bias if the patients were selected from these studies.

4.   Clear conclusion needs to be formulated in 5. Conclusions section instead of showing the scope and future perspective of this work.

Author Response

We would like to thank the reviewer for critically evaluating our manuscript and for the constructive comments and suggestions, which we used as guidance to significantly improve the quality of our manuscript. Please find as follows our point-by-point response to the reviewer's comments:

    There are several concerns as below for this manuscript.

  1. Abbreviations like PTC, NMD can be voided for easy reading

The unnecessary abbreviations are now removed from the text.

  1. The term functional consequences gives misleading interpretation that the study demonstrates impact of WT1 on malignant transformation or suril or other cellular functions. The better term could be "Understanding mutation and gene expression interactions of WT1 in adult? leukemia and lymphoma"

Thank you very much for this great suggestion, we have modified the title of the manuscript.

  1. Give a brief description of the patients included in this study from trials AML-BFM 04 and AML-BFM 2012. Clarify why patient samples were obtained as part of this trials. Justify that there was no genetic bias if the patients were selected from these studies.

A subsection including additional information regarding the patient samples were added to the methods.  We have also added another supplementary table (Table S6), describing the characteristics of these patients in more detail. Shortly summarized, from patients recruited in AML-BFM 04 and 12 trials between June 2003 and November 2017, 24 patients with WT1 alterations were randomlyselected. As described in the newly added Table S6, these patients showed characteristics comparable to our previously published cohort of 353 patients, 48 with WT1 mutations (Niktoreh et al., Journal of Oncology, 2019) where majority of the patients were diagnosed with AML M1/M2, the median age at diagnosis was 11 years and FLT3-ITD was the most common cooccurring mutation in the group of WT1 mutated patients. This information was added to the relevant section in the results. Nevertheless, we agree with the reviewer that this small cohort of 24 patients is not representative of all patients with WT1 mutations, thus we have refrained from performing critical Kaplan-Meier analyses such as survival  or relapse rates.

  1. Clear conclusion needs to be formulated in 5. Conclusions section instead of showing the scope and future perspective of this work.

The conclusion was revised as suggested, thank you.

Reviewer 3 Report

In manuscript Cancers-2411712, Niktoreh et al. study WT1 alterations in several cancer cell lines and primary blood cancer samples. In addition, they investigated alternative splicing and non-sense mediated decay of WT1 transcripts (normal and mutated).

While the topic is of interest, as a major point of criticism, the authors unfortunately did not make sufficient use of already published and readily available sequencing data for many of the cell lines. These data would allow for extension and confirmation of their newly generated results. Furthermore, the study does not really contribute much to the understanding of the functional consequences of WT1 alterations, as stated in the title. To address this, for example, it would be necessary to perform time-consuming CRISPR/Cas9 experiments that facilitate introduction/correction of WT1 alterations and investigation of their impact on leukemia/lymphoma cell proliferation/survival, gene expression etc.

Minor points:

2.2 Please declare which primer was used for RT (oligo-dT or random)?

Only one housekeeping gene was used for TaqMan normalization. Usually, ABL1 is used in combination with GUSB.

2.5 How did the authors verify that the cloned regions recapitulate the WT1 splicing activity correctly? Non-expressing U937 was used, but also a cell line expressing endogenous as well as exogenous WT1 could have been generated. Sequence specific RTs with 3’UTR and IRES primer for endogenous and exogenous WT1, respectively, followed by PCR could be employed to discern and analyze the transcriptional output of the two sources separately.

Did the authors try to quantify the splicing variants by making use of publicly available RNA-seq data sets for some of the cancer cell lines (e.g. from CCLE or DSMZ)? It would be interesting to see whether this data is in line with their PCR fragment analyses results regarding inclusion of exon 5 and KTS.

Some examples for Ki67 flow cytometry could be shown in the Supplements. Was the distribution in this case bimodal (positive proliferating versus negative resting cells), and the positive fraction increased in faster dividing lines, as to be expected?

Line 257 : typo “the its”

Line 268: if I understood correctly, three lines harboring WT1 mutations were identified by cDNA sequencing and four by NGS (Table S5). Please, explain the discrepancy.

Line 284: typo “florescent”

Figure 3: in panel B the AUCs of ABL1 and WT1 appear to be similar for +/+ (should be approx. 1 in panel C), and for universal the sum of all four WT1 variants appears larger than the ABL1 AUC (should be >1 in C). Please explain, why it is the other way round in Fig. 3.

Line 326 ff: in my view this pipeline is rather more filtering than validation. Furthermore, the targeted sequencing approach covered only known hotspot regions (except for Sanger-sequenced WT1). As already mentioned above, RNA-seq as well as exome and/or WGS data are available for many of the cell lines and should be used to confirm/validate most of the novel variants, e.g. GATA2 c.344_351delCCTGGACC in M-07e (SRR8615762 and ERR3003561 files in short read archive).

Line 527 typo: it should probably read western blot (not blood ;-).

Please, if applicable, include genetic subtypes (e.g. KMT2A fusions) in Table S1.

the quality of English is okay

Author Response

We would like to thank the reviewer for the careful evaluation of our manuscript and the on-point comments, revealing some unfortunate mistakes in the text as well as the figures. We realized that the reviewer has rightfully pointed out that the title of the manuscript is not ideally reflecting the content of the information provided in the manuscript, thus potentially leading to confusion of the reader or false expectations. As correctly mentioned by the reviewer, advanced functional experiments with primary AML blasts may be required for broader conclusions, as suggested in the title. Although especially the CRISPR/Cas9 experiments would be of great interest and are certainly worth to perform with primary AML blasts, they would require in vivo studies in immunodeficient mice, as we cannot grow the primary blasts in in vitro cultures. As these studies clearly are outside of the scope of this manuscript, we resolved this point by changing the title of the revised manuscript into a more specific option, based on the suggestions of reviewer 2. Using additional public data sets, we have also added the missing information regarding the cytogenetic background of the cell lines included in this study. Furthermore, by employing bioinformatician expertise, we have performed a deeper search to confirm our data (please refer to our answers below for more details). We believe that the addition of these information as well as the correction of the errors in the manuscript has significantly increased the quality of this manuscript. Please find as follows our point-by-point response to the reviewer's comments:

Minor points:

2.2 Please declare which primer was used for RT (oligo-dT or random)?

The RT was performed using SuperScriptTM VILO cDNA Synthesis Kit, which applies a mixture of oligo-dTs and random hexamers. This information is now added to the manuscript.

Only one housekeeping gene was used for TaqMan normalization. Usually, ABL1 is used in combination with GUSB.

We initially tested GUSB and B2M as housekeeping genes in addition to ABL1; however, to our surprise, both genes showed very high heterogeneity in expression levels across different cell lines and thus, we decided to exclude these from the final analysis.

2.5 How did the authors verify that the cloned regions recapitulate the WT1 splicing activity correctly? Non-expressing U937 was used, but also a cell line expressing endogenous as well as exogenous WT1 could have been generated. Sequence specific RTs with 3’UTR and IRES primer for endogenous and exogenous WT1, respectively, followed by PCR could be employed to discern and analyze the transcriptional output of the two sources separately.

We agree with the reviewer that in U-937 cells, silencing of WT1 is probably a result of so far undescribed epigenetic regulations. However, we believe that using a lentiviral expression system and the strong promoter of spleen focus-forming virus (SFFV), these regulatory mechanisms could be circumvented, making U-937 an even more interesting model to study WT1 isoform expression. Here, it is important to point out that the SFFV promoter (from Christopher Baum’s Lab in Hannover) has been used by several groups for stable expression of genes in murine and human hematopoietic cells in mouse models for years. Nevertheless, if the endogenous WT1 promoter is used instead of SFFV, the suggestion of the reviewer would be very helpful to measure the expression of WT1 in WT1expressing vs. non-expressing cells.

Did the authors try to quantify the splicing variants by making use of publicly available RNA-seq data sets for some of the cancer cell lines (e.g. from CCLE or DSMZ)? It would be interesting to see whether this data is in line with their PCR fragment analyses results regarding inclusion of exon 5 and KTS.

We would like to thank reviewer 3 for this great suggestion. However, the major alternative splice sites of WT1 are located on exon 4/5 and 9/10 with 483 bps in between for the longest isoform, WT1+/+. Thus, in order to extract the splicing profile of WT1 from RNA sequencing data, a depository with read lengths of at least 500 to 600 bps is needed. In DSMZ’s LL-100 project (PMID: 31160637), the RNA sequencing has been performed with read lengths of 2x151 bps. The CCLE’s latest released RNA sequencing data (PMID: 31068700) includes fragment sizes of maximum 400 bps. Other RNA sequencing data sets with longer read-length are not publicly available. Therefore, although of great interest, performing the analysis asked by the reviewer is currently not possible due to this methodological limitation.

Some examples for Ki67 flow cytometry could be shown in the Supplements. Was the distribution in this case bimodal (positive proliferating versus negative resting cells), and the positive fraction increased in faster dividing lines, as to be expected?

As suggested by the reviewer, example histograms were added to the supplementary figure S1 as Panel C. In the flow cytometry experiments performed in this manuscript, we used Ki67 expression as a control for successful permeabilization and fixation of the cells. We did not use the Ki67 expression as an additional normalization parameter.

Line 257 : typo “the its”

The typo was corrected, thank you.

Line 268: if I understood correctly, three lines harboring WT1 mutations were identified by cDNA sequencing and four by NGS (Table S5). Please, explain the discrepancy.

We noticed that a mistake has been occurred when writing this section: as correctly pointed out by the reviewer, all four cell lines with WT1 mutations, identified in both Sanger sequencing and NGS (CTS, KASUMI-6, U-937 and MOLM-1) were included in these experiments and (shown in Figure 1A). This error is now corrected in the revised manuscript.

Line 284: typo “florescent”

The typo was corrected.

Figure 3: in panel B the AUCs of ABL1 and WT1 appear to be similar for +/+ (should be approx. 1 in panel C), and for universal the sum of all four WT1 variants appears larger than the ABL1 AUC (should be >1 in C). Please explain, why it is the other way round in Fig. 3.

We would like to thank reviewer 3 for noticing this discrepancy. During the production of this figure panel, a copy error had occurred when transferring the data from the analysis software GeneMapperTM to the excel sheets and into the GraphPad! We went to the raw data and analyzed the curves again and corrected this Figure. The authors are very grateful that reviewer has noticed this inconsistency.

Line 326 ff: in my view this pipeline is rather more filtering than validation. Furthermore, the targeted sequencing approach covered only known hotspot regions (except for Sanger-sequenced WT1). As already mentioned above, RNA-seq as well as exome and/or WGS data are available for many of the cell lines and should be used to confirm/validate most of the novel variants, e.g. GATA2 c.344_351delCCTGGACC in M-07e (SRR8615762 and ERR3003561 files in short read archive).

We would like to thank reviewer 3 for noticing the missing mutation in M-07e, which regardless of several rounds of controlling the data bases did not make it into the original manuscript. To avoid any other missing variants, using bioinformatician expertise and the genomic position of our acclaimed novel variants, we performed a thorough analysis of the raw data depositories of the CCLE at https://depmap.org/portal/download/all/. Except for the mentioned GATA2 mutation in M-07e (p.Pro115ArgfsTer67), we only detected the BCOR mutation (p.Trp1663Ter) however not in KG-1 cell line but in SKLMS1 cells which is not included in our analysis. Using this additional layer of control, the data has been updated and we very much hope that no other variants have been missed.

Line 527 typo: it should probably read western blot (not blood ;-).

The typo was corrected.

Please, if applicable, include genetic subtypes (e.g. KMT2A fusions) in Table S1.

This information was added to Table S1.

Round 2

Reviewer 1 Report

OK but with modifications again.

A point is need about why it’s important such study of the biology of wt1 in terms of treatment with an additional chapter in the discussion and a sentence in the conclusion

The limitations about cell lines should be pint pointed

Author Response

Thank you for the suggestion, a new section was added to the discussion. Please note that all changes are highlighted in the text.

Reviewer 3 Report

The authors have addressed sufficiently the points I raised, and improved and corrected the manuscript. The new title also fits to the content now. After careful proof reading (e.g. typo in new text line 129 "fist diagnosis" ;-) the paper should be worth publishing.

The English is appropriate.

Author Response

We would like to thank the reviewer again for their constructive comments. We have carefully reviewed the manuscript upon the re-submission and corrected the remaining typos.